# A Non-Invasive Approach in the Assessment of Stress Phenomena and Impairment Values in Pea Seeds Caused by Pea Weevil

**DOI:** 10.3390/plants10071470

**Published:** 2021-07-18

**Authors:** Sándor Keszthelyi, Dániel Fajtai, Zsolt Pónya, Katalin Somfalvi-Tóth, Tamás Donkó

**Affiliations:** 1Department of Agronomy, Institute of Agronomy, Hungarian University of Agriculture and Life Sciences, S. Guba str 40., H-7400 Kaposvár, Hungary; ponyazs@yahoo.com; 2Medicopus Nonprofit Ltd., S. Guba str 40., H-7400 Kaposvár, Hungary; daniel.fajtai@sic.medicopus.hu (D.F.); donko.tamas@sic.medicopus.hu (T.D.); 3Department of Water Management and Climate Adaption, Institute of Environmental Science, Hungarian University of Agriculture and Life Sciences, S. Guba str 40., H-7400 Kaposvár, Hungary; somfalvi-toth.katalin@uni-mate.hu

**Keywords:** bioluminescence imaging, *Bruchus pisorum*, CT analysis, IR thermography, non-invasive approach, *Pisum sativum*, plant stress, seed impairment

## Abstract

Pea (*Pisum sativum* L.) is an important leguminous plant worldwide, in which pests trigger significant damage every year. One of the most important pest is pea weevil (*Bruchus pisorum*, L) which causes covert damage in crops. In the present study, our aim was to obtain precise information pertaining to the extent and the nature of damage in pea caused by *B. pisorum* by means of non-invasive imaging methods. The infested pea samples were analysed by an infrared thermometer and a bioluminescence plant imaging system as well as a computer tomograph under laboratory conditions. The calculated weight of organic matter destroyed by the developing larvae was 36.46%. The changing of RGB (red, blue, green) codes obtained through thermal imaging and the CPS (counts per second) values originating from bioluminescence imaging in infested samples were statistically verifiable. According to our CT assay, the damage caused by *B. pisorum* changed the tissue density, volume and shape of the pea seeds by the end of the development of the pest. The results of thermal and bioluminescence imaging contribute to a better understanding of the internal chemical processes and the CT analysis helps to understand the alteration trends of the inner structure of seeds caused by this pest.

## 1. Introduction

Pea (*Pisum sativum* L.) is an important leguminous plant in both arable and horticultural farming. Its considerable protein, vitamin and mineral content play a crucial role in human and animal nutrition [1]. The content of the organic matter of the seed is influenced by several biotic and abiotic stress factors causing a 31.9 to 69.6% loss in crop productivity on global average due to various pests [2]. The phytophagous arthropods can cause damage both in the field and under post-harvest conditions. Insect pests alone are responsible for about 20% of all damage in leguminous crops [3].

Seed quality is one of the most important features of a successful harvest, whose decline can be traced to the presence of phytophagous pests. A fast deterioration in seed quality can occur due to the activity of stored products pests, which is reflected by the deterioration of seed viability and vigour [4]. This deterioration caused in quality by pests can be related to various changes in physiological processes at the cellular, metabolic and biochemical levels such as erroneous RNA synthesis, protein synthesis breakdown, lipid peroxidation, membrane disruption and damage brought about by free radicals, which causes several detrimental effects on the seed [5]. The detection of these stresses can contribute to the containment of infestation and to effective protection practices.

One of the major pests in pea crops worldwide is the pea weevil (*Bruchus pisorum* L.) (Coleoptera: Chrysomelidae) [6]. It has become a typical cosmopolitan pest in the last century [7]. The damage is mainly caused by the developing larvae, which appear during storage [8]. Adult weevils are present in pea crops after their wintering diapause termination in early spring. Single eggs are laid by females on the surface of pea pods [9]. The hatched larvae chew through the pod wall, where it develops alone inside the soft seed [10]. The larvae can destroy more than 50% of the cotyledon in pea grains, leading to empty grains, poor seed germination and poor seed quality [11]. The economic damage triggered by *B. pisorum* varies considerably, and infestation levels reported worldwide range from 10% to 90% [12].

Methods suitable to yield an objective assessment of damage brought about by the most important covertly developing pests in their hosts have not been explored as yet. These evaluations are mainly based on traditional, invasive methods, e.g., weighing, dissections of damaged plants/plant parts or performing wet chemical analysis on plant structures [13,14]. At present, many novel imaging technologies such as computer tomography, magnetic resonance or bioluminescence imaging [15,16,17] are known, which can be exploited to precisely determine the level and degree of injuries in an attacked plant. Albeit these methods are scarcely used in plant sciences [18], there are several studies [19,20,21] that have reported radically new results obtained by using these novel techniques.

An objective picture of the damage caused by several hidden arthropod pests, and of their biological traits, without disturbing their living space, can only be attained by employing non-invasive methods, which are rather uncommon even in modern plant biology research. Therefore, our aim was to obtain detailed information via non-destructive imaging technologies about the extent and the nature of damage in pea caused by *B. pisorum*. Based on our findings, we urge the use of these methods in monitoring the damage of seeds during storage. Taken together, our results can contribute to preventing damage of seeds in the course of storage.

## 2. Results

The average infestation rate of the experimental pea sample was 21%. The weights of the healthy (1683.4 ± 59.7 mg) and damaged pea seeds (1069.7 ± 47.9 mg) were significantly different, as had been expected (df = 1; *p* < 0.001). The calculated weight of organic matter destroyed by the developing larvae was 613.7 ± 11.7 mg, which represents 36.46% weight loss in the examined leguminous items.

The results of the IR thermal imaging of healthy pea seeds and those damaged by *B. pisorum* are shown in Figure 1. The alteration of colour between the healthy *vs*. infested samples was conspicuous 24 h after germination. The darker, bluish-green shades of the damaged seeds reflect their lower inner temperature.

The result of the red, green, blue (RGB) colour analysis revealed that there was no significant difference in IR images (i.e., in the heat radiation) of healthy and damaged pea seeds. Figure 2 shows the 3D scatterplot of RGB codes (Red, Green, Blue codes ranging from 0 to 255) of healthy (a) and damaged (b) pea seeds. The one-way ANOVA showed that the p-values of Red, Green, Blue codes between healthy and damaged seeds were *p* < 0.001, 0.186, and *p* < 0.001, respectively. The calculated F-value in the cases of red and blue codes were 42.73 and 173.63, respectively, while the critical F-value with n-2 freedom of degree and on 0.05 significance level was 3.84 in both cases. The calculated value exceeds the critical value. The F-value of green codes was 1.75 with the same critical F-value. Therefore, it can be stated, that there was no significant difference between the red and blue codes, while there was a significant difference between the green codes of healthy and damaged pea weeds.

Statistically justifiable difference (df = 1, *p* = 0.048) between the level of biophoton emission of healthy samples and that of the infested seeds could be detected (Figure 3), revealing a lower emission in the latter. The overall counts per second (CPS) values in the healthy samples reflected a rather “levelled” UPE emission—probably representing a “baseline”, spontaneous autoluminescence—except for two emission peaks (sample “M 54” and “M 76”).

The kernel tissue densities (HU) measured by computed tomography of healthy and damaged peas can be seen in Table 1. The healthy samples had a much higher radio-density than damaged samples; the difference being statistically verifiable (df = 1; *p* < 0.001). The mean changing density of the damaged pea seeds was significant, 136,416 HU and 41.93%. Finally, the dense tissue structure had been destroyed in the pea kernels, the cause of which may be due to mastication by *B. pisorum*.

The substantial volume decrease in the organic matter in damaged peas measured by computer tomography (CT) could be statistically confirmed (df = 1, *p* =1.5 × 10^10^). The volume of the examined intact samples was 127.72 ± 4.565 mm^3^. The decrease in the volume of the damaged samples was 43.77 mm^3^ on average, (34.27% change). The CT post-processing analysis indicated that the impaired seeds suffered a 39.065 m^3^ (31.757%) volume loss.

The relative decrease in sphericity (RSph) values in damaged samples were also statistically significant (df = 1, *p* < 0.001). The average difference between the healthy and damaged seeds was 36.10%.

CT-based 3D-rendered images of pea kernels damaged by *B. pisorum* can be seen in Figure 4. The pericarp remained intact in most cases, however emergence holes of adults were conspicuous. The destruction of the centrally situated cotyledon was detectable based on the location of visible cavities explored by CT. The reserve nutrient content necessary for the development of young plants was entirely destroyed by the larvae.

## 3. Discussion

According to our results, the RGB colour analysis originating from IR thermal imaging reflects “pre-emptively” the orientation of the inner impairment of pea seeds caused by the hidden pest. Our results confirm the findings of Tuhid et al. [22], who analysed RGB components stemming from plant surfaces in order to differentiate between orchids infected by phytopathogen, and their findings showed that the proposed method is capable of detecting the disease as well as some diseases categorised into different groups. The enzymatic processes involving heat production by the seed [23] as well as its differences caused by stressors could be detected by IR thermal imaging. Obviously, the method is not suitable for investigating biological details taking place in the background, it only allows indirect inference from ensuing physiological abnormalities.

The difference in the degree of biophoton emission between the control vs. infested samples suggests that the extent to which the seed population was damaged surpassed a certain threshold of intactness, hence hampering the “normal” unfolding of those stress-related processes occurring in the tissues, which are widely believed to be behind enhanced bioluminescence induced by stress. 

The detected emission peaks may reflect the differences in the oxidative metabolic state that are probably attributable to the epigenetic pattern of the female gametophyton, reflecting individual differences in the analysed seed stock [24]. It is important to note, however, that reactive oxygen species (ROS) are the inherent “by-product” of oxidative metabolism and they play a pivotal role in signalling during seed development [25] as well, thus, their presence and their excited state cannot exclusively be attributed to stress-related cellular responses.

According to the results of our CT assay, the damage caused by *B. pisorum* changed the tissue density, volume and shape (RSph) of the pea seeds by the time the development of the pest was completed. The total weight loss of stored leguminous seeds could even reach 40% [26]. The level of impairment observed in this study was found to be close to this reported value.

The decrease in tissue density in damaged peas can be explained by the consumption of dense reserve components (starch, protein, herbal oil) in the cotyledon [27,28]. Naturally, stored product pests may also damage the embryos, causing a decrease in protein level and starch content [29], which was indirectly confirmed by both CT and IR thermal analysis.

## 4. Materials and Methods

### 4.1. Sampling

To determine the different impairment values and seed purity of pea seeds caused by *B. pisorum,* damaged pea samples were collected from a warehouse. The pea seeds, which originated from the harvest of the previous year, were stored in bags placed in a chamber with 13.5–14.0% storage moisture. The infestation began in the field and the damaging process continued with larval development. The place of the collection was Babócsa (Somogy county, Hungary) on the 15th of February, 2021. The seed moisture was determined using a Memmert UFE 400–800 heat oven (Memmert GmbH and Co KG., Schwabach, Germany), with official methodology [30]. The weight of samples was measured by an Ohaus Explorer Pro EP214CE device. Subsequently, 40 healthy and 40 damaged seeds infested by *B. pisorum* were isolated based on the visible holes made by the larvae on the surface of the seed coat.

### 4.2. Thermal Imaging and Data Processing

During IR imaging, 2 × 20 healthy and 2 × 20 damaged seeds were placed in Petri dishes between filter papers, which were soaked in distilled water for 2 h. Subsequently, the water was decanted and the dishes were covered again with lids during germination while put into an incubator at 24 °C and 75 rh [31].

IR images were taken at 24 h after the commencement of germination, during which the lids were removed in order to avoid light reflection. The distance between the thermal camera and the dishes containing the seeds was uniformly 30 cm. The IR equipment employed was a Flir I3 thermal camera (63906–0604, Teledyne FLIR Company, Calif. and Arlington, USA) with a field of view (FOV) of: 12.5° × 12.5° and a thermal sensitivity (N.E.T.D) of < 0.15 °C/150 mK producing thermal images with a resolution of 3600 pixels (60 × 60 pixels) with high-speed data acquisition real-time 14-bit digital output. Storage and analysis of the infrared images were compatible with the ThermaCAM Researcher software (Flir).

The IR images were analysed using the R statistical software “knitr” library. At first, the pea seeds were identified in the IR images and isolated by determining the gradients of the colour scale at the edges of pea seeds. Secondly, the RGB codes of each pixel of pea seeds were determined and analysed. One-way ANOVA was applied to study the statistical relationship between healthy and damaged seeds (*p* ≤ 0.05) as independent, as well as RGB codes as dependent variables. To confirm the independence, a Fisher’s F-test was performed in connection with RGB analysis of thermal imaging. It was assumed that the data had normal distribution N (m, σ2) with m expected value and σ2 variance. The null hypothesis (H0) supposed that there was no significant difference between the RGB codes of healthy and damaged pea seeds.

### 4.3. Bioluminescence Imaging

Pea seeds were imaged by employing the NightShade LB 985 Plant Imaging System (Berthold Technologies, Bad Wildbad, Germany) equipped with a highly sensitive CCD-sensor (NightOWLcam, Berthold Technologies) thermoelectrically-cooled to −74 °C to minimise thermal noise. The camera was mounted on a dark, light-tight chamber into which the seeds to be imaged were placed (approximately 20 cm away from the camera lens). Luminescence emissions from the control seeds and weevil-treated seeds were imaged and photons emanating from the samples were detected with a back-lit, midband-coated full frame chip possessing a spectral range of 350–1050 nm (with quantum efficiency of 90% at 620 nm wavelength). During image acquisition the binning factor was set to: 2 × 2 via the software, hence, the images were captured at the resolution of 512 × 512 pixels at a final 26 × 26 µm² pixel size (slow scan mode). The exposure time was set to 60 s. For image analysis the IndiGo software (V. 2.0.5.0, Berthold Technologies, Germany) was used. Before each measurement, all the samples were kept in the light-tight dark chamber for 1 h.

The CPS values of healthy and damaged samples caused by *B. pisorum* were examined statistically by one-way ANOVA. Means were separated by using the Tukey (HSD) test, at *p* ≤ 0.05.

### 4.4. CT Acquisition and Image Post-Processing

Computed tomography (CT) data collection was performed by Siemens Somatom Definition AS+ (Siemens, Erlangen, Germany) scanner using the settings as follows: tube voltage 100 kV, current 175 mAs, spiral data collection with pitch factor 0.7, collimation 128 × 0.6 mm. CT images were reconstructed of 40 “control” (C) and 40 “damaged” (D) peas arranged in a grid-like structure (8 rows, 5 columns) with 51 mm Field of View (voxel size of 0.996 × 0.996 × 0.1 mm).

The stored images in DICOM (Digital Imaging and Communications in Medicine) format were converted to NifTI (Neuroimaging Informatics Technology Initiative) format; then, every image processing step was performed in Python with open-source or custom-made software library and code. Otsu’s thresholding method was used in combination with morphological operators for producing initial binary masks (M1), which represents the actual state of the peas. Masks describing the approximated intact condition of the peas (M2) were created from M1 with additional “growing” morphological operators—thus, the voxels of M1 were a subset of M2’s voxels (Figure 5).

The mean and the standard deviation values of radiodensity were calculated for every pea on the images using the individual M1 and M2 masks. The sphericity [32] of the M1 and M2 masks of every pea were determined using the surface of the triangle mesh generated with the marching cubes algorithm. The damages and shape abnormalities of the peas were characterized by ratio of the sphericity values of the individual M1 and M2 masks. The relative sphericity (RSph) was calculated (RSph = sphericity(M2)/sphericity(M1) × 100), from which we were able to draw conclusions about the shape of the pea seeds after the insect damage.

Density (HU = Hounsfield Unit) and volume (mm^3^) values of solid seed constituents, as well as the relative sphericity of the intact and damaged samples, were statistically analysed by one-way ANOVA by using the SPSS for Windows 11.5. software package. Mean values were separated by using the Tukey (HSD) test, at *p* ≤ 0.05.

## 5. Conclusions

Our non-invasive experimental investigation provides new data about the consequences of stress in germinating pea seeds triggered by an insect pest, such as visual imaging of the vitality and degree of stress phenomenon of the seeds as well as cavitation formation and tissue density changing inside the seed. Furthermore, the obtained data represent a novel approach in the study of the physiology of pea seeds and its hidden lifestyle pests. This physiological response of the host was reflected by the lower inside temperature in impaired seeds caused by decreasing enzyme activities or the higher bioluminescence light emission derived from the increase in reactive oxygen species.

The results of thermal and bioluminescence imaging contribute to a more profound understanding of the inner chemical phenomena occurring in infested seeds. Additionally, CT analysis helps to understand the alteration trends of the hidden structure of seeds caused by an insect pest. Our novel findings, associated with relative sphericity, have unequivocally pointed out pea shape alteration caused by insect damage, which may facilitate the development of an artificial pre-sorting method for damaged seeds.

Among the applied methods, the data provided by the IR thermography are the easiest to produce and access, which is due to the compactness and handy employment of the IR thermometer. This offers an economical and ideal, non-destructive diagnostic method for pest detection, which is suitable for on-the-spot inspection of stored products. Bioluminescence imaging only allows for the quick inspection of smaller samples transported to the laboratory. In contrast, CT-assisted diagnoses have provided a more comprehensive and more in-depth image of the features of the hidden damage. Undoubtedly, however, this method is more time-consuming, the instrument is expensive and requires thorough expertise to use, as compared to other methods discussed above.

In summary, the non-invasive methods employed in this study can provide additional data about the physiological conditions of seeds as well as biological and ecological information on the lifestyle of covertly developing insects, which has been largely unknown. Thereby, our results can greatly contribute to efforts aimed at stress alleviation and to their successful implementation through the early detection of the presence of hidden seed pests in seed stocks. Based on our results, further elaboration of our approach, and the combined use of imaging instruments, can facilitate the development of novel industrial protection methods against stored product pests during the post-harvest processes.

## Figures and Tables

**Figure 1 plants-10-01470-f001:**
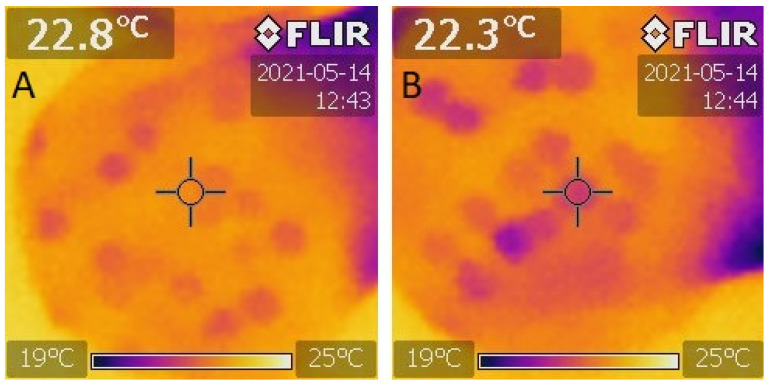
IR thermal imaging of healthy and damaged pea seeds caused by *B. pisorum* 24 h after artificial germination. (**A**): healthy; (**B**): damaged seed.

**Figure 2 plants-10-01470-f002:**
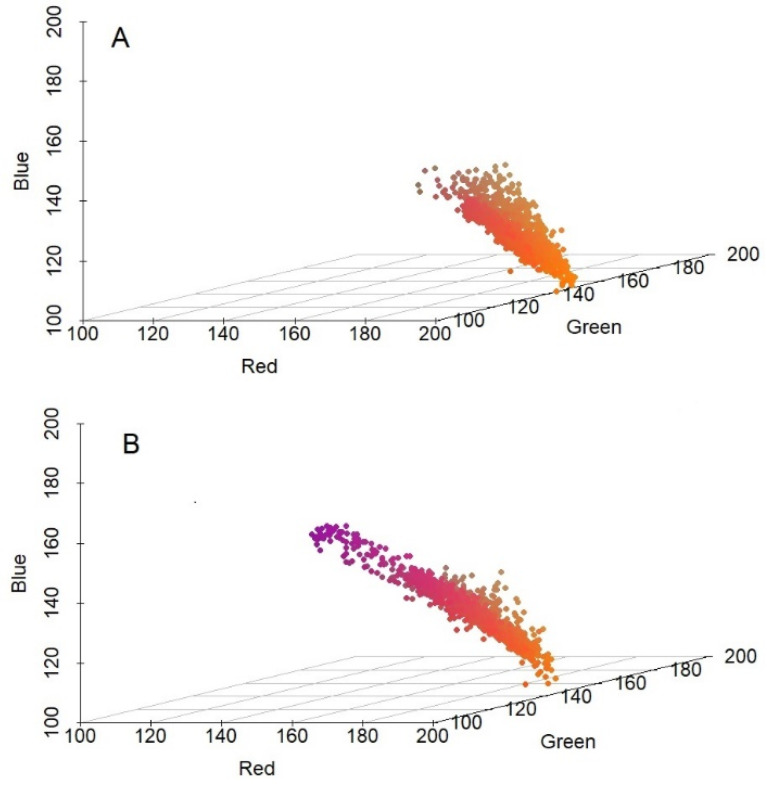
IR thermal imaging of healthy and damaged pea seeds caused by *B. pisorum* 24 h after the artificial germination. (**A**): healthy; (**B**): damaged seeds.

**Figure 3 plants-10-01470-f003:**
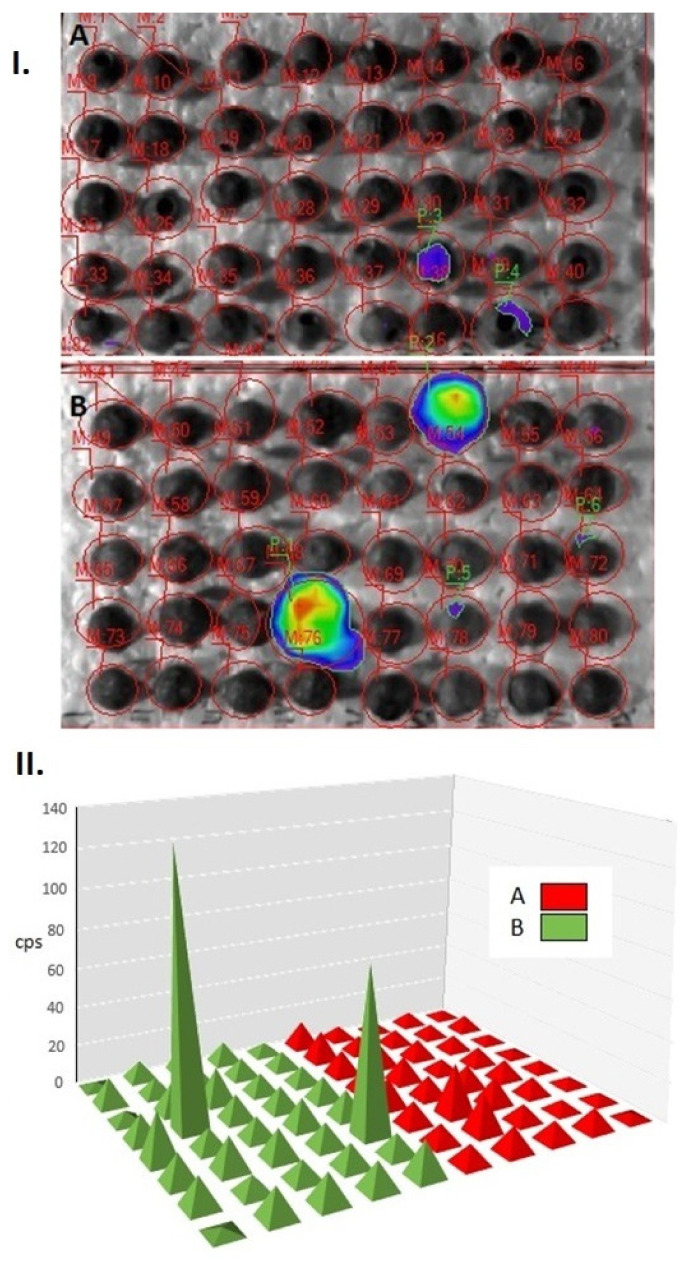
Data of biophoton emission (average counts per second) recorded from pea seeds (**I**). 2 D images of pseudo colour-coded pixel intensity values; (**II**). CPS values illustrated in coordinate system; A: healthy; B: damaged seeds.

**Figure 4 plants-10-01470-f004:**
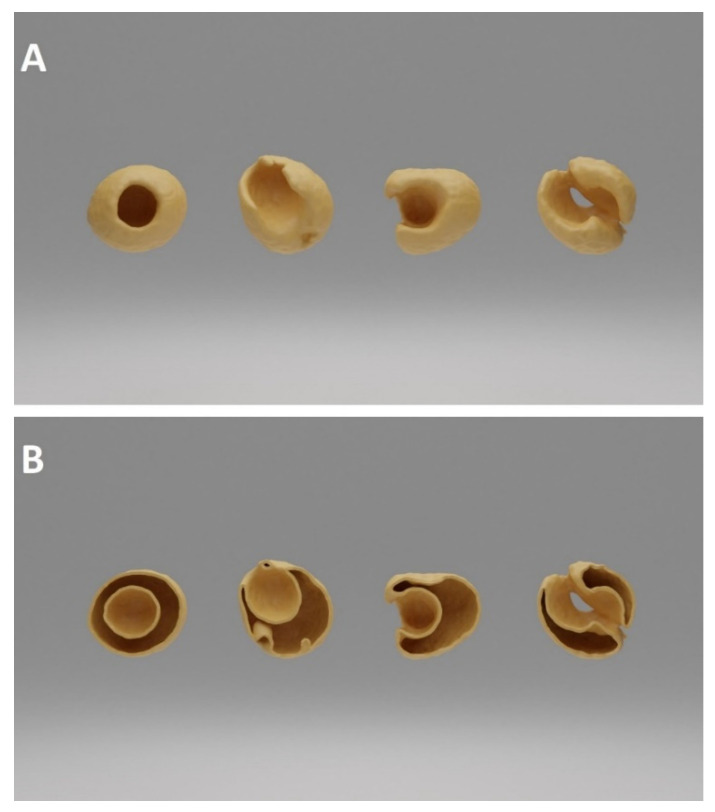
Computed tomography-based 3D-rendered image of the cavity triggered by *B. pisorum* larvae in pea seeds. (**A**): volume illustration of damaged seed; (**B**): surface illustration of a damaged seed.

**Figure 5 plants-10-01470-f005:**
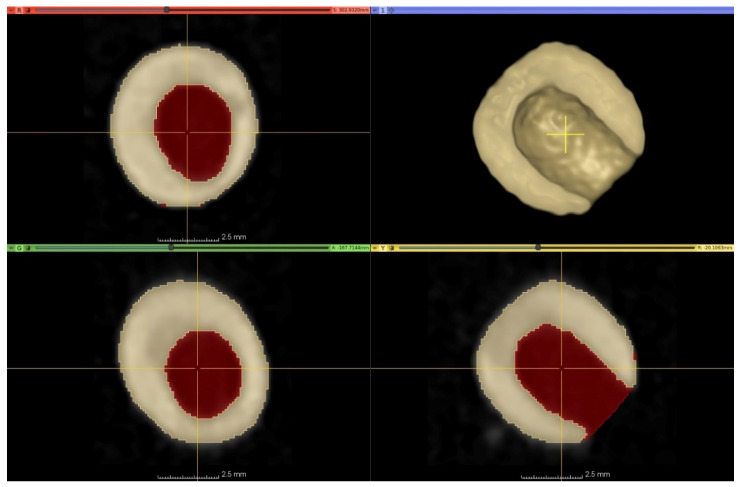
Three-plane cross-sectional image and 3D-rendered representation of damaged peas by CT. The area of the lesion is marked in red in the cross-sectional images. The initial binary mask representing the actual volume of a pea (M1) marked with yellowish colour in the cross-sectional and the 3D-rendered image. The area of the lesion is marked with the reddish colour in the cross-sectional images. The mask approximating the intact condition of the peas (M2) is the union of the yellowish and the reddish areas on the cross-sectional images.

**Table 1 plants-10-01470-t001:** Statistical data of CT-assisted density, volume and relative sphericity values of healthy and damaged pea samples by *B. pisorum* (*n* = 40); HU = Hounsfield Unit; RSph = relative sphericity (*p* < 0.001).

		Mean	std.dev.	SE	Variance
density (HU)	intact	170.127	30.482	4.823	929.193
	damaged	33.710	52.725	8.342	2779.981
volume (mm^3^)	healthy	127.729	28.852	4.565	832.490
	damaged	83.951	24.001	3.797	576.006
RSph	healthy	95.572	6.712	1.062	45.062
	damaged	61.069	10.054	1.590	101.098

## Data Availability

Data available on request.

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
