# Peer review of "A Non-Invasive Approach in the Assessment of Stress Phenomena and Impairment Values in Pea Seeds Caused by Pea Weevil"

_plants, 2021, doi:10.3390/plants10071470_

Round 1
Reviewer 1 Report
Keszthelyi et al. describe the application of spectroscopic methods for the non-invasive analysis of pea seeds. The authors also claim new insights into the physiological process of infested seeds.
However, the paper in the present form lacks in
A) explaining well the outreach of the applied techniques. I.e., Which are the parameters that can be measured with each method? What are their throughput and their cost? Are they fieldable?
B) giving new information about insect pest induced changes in the seed; ideally, on a chemical level, and time-resolved.
C) providing recommendations on the use of such methods to monitor storage damages of seeds.
Before the eventual publication of the manuscript, the authors should redefine their scope and restructure their text to present a consistent and convincing story.
Abstract
Both the abstract and the conclusions refer to a new approach for studying the physiological processes in seeds. However, there are no biochemical data presented in the manuscript.
Discussion
"This statement was confirmed by the result of Tuhid et al. [20],..." does not make sense since Tuhid et al. obviously published first. "Our results confirm the findings of..." would better describe the situation.
The interpretation of results is somehow speculative. For example, lines 151-163 would need support. But, at least the ROS could be determined easily.
Besides, the results seem to be over-interpreted concerning the population's genetic structure (164-169) since commercial material (no supporting genomic data) was used and only a single sample with relatively few seeds.
Methods
- How did the authors control (temperature, humidity) and measure the seed damage? Traditional methods are, for example, the production of flour and weight loss.
- Why didn't the authors use data mining methods, such as Ada Boost and Random Forest? They probably would work great for classification.
Code and data availability
The authors should make their data and scripts (Python/ R code) available for review and interested readers.
Conclusions
Again, the text could be more concise. For example,
"provides new data"..."novel approach" is not contributing information for the reader.
Altogether, the conclusion remains too vague. It should
- Detail about the physiological changes that can be detected with the presented non-invasive methods.
- Give some information, which of these techniques has shown the best analytical performance and the best cost-benefit.
- Explain possible applications to prevent storage damage of seeds.
- What would be the advantage of such methods against simple weighting?
Writing
1. Several expressions seem not to fit, such as:
-pea items
-field circumstances
-organic matter... is endangered (referring to proteins, vitamins, etc.)
-The damage is triggered
-most important hidden lifestyle pests
-...
The text is understandable, but a native speaker should revise the wording to avoid misinterpretations.
2. Besides, the style could be optimized, for example, the phrase:
"The economic damage triggered by B. pisorum often reaches a considerable degree. Nevertheless, the exact data of yield loss caused by it varies considerably and infestation levels reported worldwide range from 10% to 90% [10]."
is wordy and could be written as :
"The economic damage triggered by B. pisorum varies considerably, and infestation levels reported worldwide range from 10% to 90% [10]."
Note: these are just examples to demonstrate that English proofreading is required.
Author Response
Dear Reviewer 1,
It gives us pleasure to resubmit our manuscript to MDPI Plants, entitled “A non-invasive approach in the assessment of stress phenomena and impairment values in pea seeds caused by pea weevil”. We appreciate the reviewer’s affirmation of this study and suggestions on the structure and organization of the writing in the previous manuscript.
We have inserted the asked thoughts into the text and we have fully revised the manuscript according to their comments. We attached the files (revised and with tracked changes) via ms center:
Our detailed comments are the followings item by item:
Explaining well the outreach of the applied techniques. I.e., Which are the parameters that can be measured with each method? What are their throughput and their cost? Are they fieldable? Giving new information about insect pest induced changes in the seed; ideally, on a chemical level, and time-resolved. Providing recommendations on the use of such methods to monitor storage damages of seeds.
Our response: It is answered in the conclusions
Before the eventual publication of the manuscript, the authors should redefine their scope and restructure their text to present a consistent and convincing story.
Our response: Our objectives were completed based on the suggestion of Rev. The affected chapters of MS were restructured.
ABSTRACT: Both the abstract and the conclusions refer to a new approach for studying the physiological processes in seeds. However, there are no biochemical data presented in the manuscript.
Our response: Indeed, our non-invasive analysis did not cover biochemical analysis. Our objectives were only the applications and comparison of some biophysical imaging methods in the pest diagnostic and its damage evaluation as a novel approach
DISCUSSION: "This statement was confirmed by the result of Tuhid et al. [20],..." does not make sense since Tuhid et al. obviously published first. "Our results confirm the findings of..." would better describe the situation.
Our response: The suggested revision was added into the text.
The interpretation of results is somehow speculative. For example, lines 151-163 would need support. But, at least the ROS could be determined easily
Our response: The mentioned paragraph was restructured and rephrased.
The results seem to be over-interpreted concerning the population's genetic structure (164-169) since commercial material (no supporting genomic data) was used and only a single sample with relatively few seeds.
Our response: This paragraph was deleted.
METHODS: How did the authors control (temperature, humidity) and measure the seed damage? Traditional methods are, for example, the production of flour and weight loss.
Our response: Our aims were the application of imaging methods and the evaluation of their utilization. Therefore, we wouldn’t like to analyse of any traditional methods.
We have measured the weight loss triggered by pea weevil, which thoughts derived from this assay are interpreted in the first paragraph of results
Why didn't the authors use data mining methods, such as Ada Boost and Random Forest? They probably would work great for classification.
Our response: AdaBoost and Random Forest methods are used to classification and decision making, but there are a few cases when their reliability decreases. They basically only work on tabular data, and we mostly have worked with images. There were only two methods that might allow to apply Ada Boost and Random Forest (i.e. RGB codes and biophoton emission intensity data), but the focus of the study was to introduce different non-invasive imaging methods in practice that can potentially be able to distinguish different stress phenomena and impairment values. To do this, basic statistical methods were usable and gave correct results.
CODE AND DATA AVAILABILITY: The authors should make their data and scripts (Python/ R code) available for review and interested readers.
Our response: The supplementary file containing the research data and processing is uploaded.
CONCLUSIONS: "provides new data"..."novel approach" is not contributing information for the reader.
Our response: The acquired new benefits of our non-invasive approaches were added.
Detail about the physiological changes that can be detected with the presented non-invasive methods.
Our response: The host physiological changes and their triggers were added.
Give some information, which of these techniques has shown the best analytical performance and the best cost-benefit. Explain possible applications to prevent storage damage of seeds. What would be the advantage of such methods against simple weighting?
Our response: The asked information and features of these discussed methods was inserted.
WRITING: Several expressions seem not to fit, such as: pea items; field circumstances, organic matter... is endangered (referring to proteins, vitamins, etc.), the damage is triggered, most important hidden lifestyle pest, The economic damage triggered by B. pisorum varies considerably, and infestation levels reported worldwide range from 10% to 90% [10]."
Our response: The mentioned mistakes were corrected. In general, the whole MS was linguistically improved by a native English speaker. (but these linguistically improvements can be seen only in the final, uploaded version)
We hope our revised MS will meet the requirements of MDPI Plants.
Thank you very much for your contribution and assistance!
Yours sincerely,
Sándor Keszthelyi

Reviewer 2 Report
The authors of this manuscript have carried out a considerable amount of work and have generated interesting results using non-destructive imaging technologies to assess the extent and the nature of the damage caused to pea seeds by the pea weevil, Bruchus pisorum, the major insect pest of pea. The findings are of considerable significance, as pea weevils are highly destructive and cause significant quantitative and qualitative losses. Moreover, in contrast to traditional, destructive methods that are commonly used to estimate the damage due to insect infestations, the methods proposed have not been previously extensively evaluated. The methods used are properly executed and adequately described in the manuscript, whereas the results are analyzed in a proper way. Some specific comments (e.g. editorial comments etc.) on the manuscript are provided on the attached annotated pdf file. In conclusion, according to my opinion, the present manuscript can be accepted for publication in Plants after a minor revision of the manuscript in which the authors will address the abovementioned comments.

Author Response
Dear Reviewer 2,
thank you for the consideration of our MS, entitled “A non-invasive approach in the assessment of stress phenomena and impairment values in pea seeds caused by pea weevil”, as well as your appreciated opinion and suggestions. We send our revised article based on the opinion of peer-reviewer 2.
We corrected all remarks, mistakes in MS, which are indicated by coloured (red) text using track-changes.
We hope our revised MS will meet the requirements of MDPI Plants.
Thank you very much for your contribution and assistance!
Yours sincerely,
Sándor Keszthelyi

Reviewer 3 Report
plants-1271355-review
The authors presented a method for evaluation of the stress affecting the pea seeds after insect infestation. The evaluation took place using three sensors.
The work looks innovative but needs some work to be publishable in the Plants Journal.
Abstract
The abbreviations of CPS and CT and any subsequent names need to be listed next to the abbreviated names at the first instance.
What does RSph stand for?
Introduction
The authors state an estimation of the destroyed organic matter by the larvae. While that seems logic to state, it doesn’t affect the decision that the pea is already damaged totally and not available for consumption. Thus, whether the damage of the seed is small or large, the seed at the end is considered not valid and needs to be discarded. This is more like a binary classification. If there is an infested seeds, they can’t be used for human consumption either as a fresh produce or processed product.
The authors need to explain how stress is reflected in seeds especially due to insect infestation.
The authors need to be clear in their objective. The methods presented are applicable for offline purposes. This should be stated. At the end, is this work transferrable to the industry? Please state the purpose clearly.
Results
It would be helpful if an image or images of the different stages of infestation are listed in the study. Additionally, samples
Figure 1. A. should be healthy not intact as they are both intact. Please the same way in similar places in the manuscript either in figures or tables captions on in the text.
In Table 1, please add significance levels between the different mean values.
In Figure 4. It would be better to add sample images for healthy peas for comparison.
Discussion
The authors need to state with justification which non-invasive sensor can be better applied to track the stress of the pea seeds after infestation.
Materials and Methods
The authors need to state the storage conditions of the samples in the warehouse.
The authors need to state the official methodology, with a reference, for moisture measurement.
Any device should be described as (model, company, city, country).
Why was the samples put between filter papers for IT imaging? Please justify with references.
It will be good to have a schematic diagram for the IR, Bioluminescence imaging, and CT imaging.
What were the independents variables for the ANOVA test for the IR data? Please make it clear.
What is the CPS data? Please add more details.
The authors change the name of healthy to intact and then control. Please stick to one only through the whole manuscript.
The authors need to state that they analysed the sliced CT images.
The author need to provide samples images for mask and cmask.
The authors state that the cmask is the mask of the intact condition of the peas. Whereas, the mask is the initial binary image. Please add more details or images as this seems vague.
What is the difference between the three images in Figure 5? Are these from different samples? Please add more details either in the figure caption or the text.
The authors didn’t put enough information on how they extracted the density and volume from each damaged image.
Conclusions
Again, the authors did not recommend a certain one sensors outperforming the other two sensor used.
Author Response
Dear Reviewer 3,
It gives us pleasure to resubmit our manuscript to MDPI Plants, entitled “A non-invasive approach in the assessment of stress phenomena and impairment values in pea seeds caused by pea weevil”. We appreciate the reviewer’s affirmation of this study and suggestions on the structure and organization of the writing in the previous manuscript.
We have inserted the asked thoughts into the text and we have fully revised the manuscript according to their comments. We attached the files (revised and with tracked changes) via ms centre:
Our detailed comments are the followings item by item:
The abbreviations of CPS and CT and any subsequent names need to be listed next to the abbreviated names at the first instance.
What does RSph stand for?
Our response: The subsequent names were displayed in the first instance, such as relative sphericity (RSph).
INTRODUCTION: The authors state an estimation of the destroyed organic matter by the larvae. While that seems logic to state, it doesn’t affect the decision that the pea is already damaged totally and not available for consumption. Thus, whether the damage of the seed is small or large, the seed at the end is considered not valid and needs to be discarded. This is more like a binary classification. If there is an infested seeds, they can’t be used for human consumption either as a fresh produce or processed product.
The authors need to explain how stress is reflected in seeds especially due to insect infestation.
Our response: One chapter related with physiological seed impairment caused by insect pest was added in introduction with two citations.
The authors need to be clear in their objective. The methods presented are applicable for offline purposes. This should be stated. At the end, is this work transferrable to the industry? Please state the purpose clearly.
Our response: The detailed aims and the industrial consequences were added both at the end of the introduction and at the end of the discussion.
RESULTS: It would be helpful if an image or images of the different stages of infestation are listed in the study. Additionally, samples.
Our response: Unfortunately, we have not such image series, because our aim was not the qualification of the damage degrees. So, we cannot add them.
Figure 1. A. should be healthy not intact as they are both intact. Please the same way in similar places in the manuscript either in figures or tables captions on in the text.
Our response: The mentioned word was replaced in the whole manuscript
In Table 1, please add significance levels between the different mean values.
Our response: The significance level (p<0.001) was added to the title of the table
In Figure 4. It would be better to add sample images for healthy peas for comparison.
Our response: Unfortunately, CT images about the healthy seeds have not made because our aim was the displaying of cavity triggered by weevil larvae. We have only numeric data about the results of CT analysis of healthy seeds. We think so the 3D rendered imaging of healthy seeds cannot show spectacular and interesting data.
DISCUSSION: The authors need to state with justification which non-invasive sensor can be better applied to track the stress of the pea seeds after infestation.
Our response: The asked chapter was added in discussion
MATERIALS AND METHODS: The authors need to state the storage conditions of the samples in the warehouse.
Our response: Storage conditions were added.
The authors need to state the official methodology, with a reference, for moisture measurement. Any device should be described as (model, company, city, country).
Our response: The asked data were added.
Why was the samples put between filter papers for IT imaging? Please justify with references.
Our response: The samples put between filter papers in order to initialize the germination as well as the beginning of the enzymatic activity. The asked reference was added.
What were the independents variables for the ANOVA test for the IR data? Please make it clear.
Our response: The asked supplementations were added.
What is the CPS data? Please add more details.
Our response: The entire name of cps was added in the first instance. CPS is a common unit of measurement in imaging that provides more information anywhere.
The authors change the name of healthy to intact and then control. Please stick to one only through the whole manuscript.
Our response: These words were replaced in whole manuscript.
The authors need to state that they analysed the sliced CT images. The author need to provide samples images for mask and cmask. The authors state that the cmask is the mask of the intact condition of the peas. Whereas, the mask is the initial binary image. Please add more details or images as this seems vague.
Our response: The asked supplementations were added to the MM.
What is the difference between the three images in Figure 5? Are these from different samples? Please add more details either in the figure caption or the text.
Our response: This is the same sample in three plane and 3D rendered way. The asked data were added.
The authors didn’t put enough information on how they extracted the density and volume from each damaged image.
Our response: The mean and the standard deviation values of the radiodensity were calculated for every peas on the images using the individual M1 and M2 masks. The sphericity [28] of the M1 and M2 masks of every pea were determined using the surface of the triangle mesh generated with the marching cubes algorithm. The damages and shape abnormalities of the peas can be characterized by ratio of the sphericity values of the individual M1 and M2 masks. The relative sphericity (RSph) was calculated (RSph=sphericity(M2)/sphericity(M1)×100) from which we were able to draw conclusions about the shape of the pea seeds after the insect damaging.
CONCLUSIONS: Again, the authors did not recommend a certain one sensors outperforming the other two sensor used.
Our response: The asked chapter was added in conclusions
We hope our revised MS will meet the requirements of MDPI Plants.
Thank you very much for your contribution and assistance!
Yours sincerely,
Sándor Keszthelyi

Round 2
Reviewer 1 Report
My main concerns that I expressed for the previous submission have not been addressed:
- It is unclear, how the damage of kernels was quantified.
- There is no new insight to biochemical processes of stress response as claimed.
- Possible effects such as ROS activation were not experimentally confirmed.
- There is no comparison of the presented method with conventional techniques for detection of seed damages.
Thus, I recommend the rejection of the manuscript. Before resubmission, the methodology and the contribution of the paper should be well explained.
Reviewer 3 Report
The authors addressed the comments in the possible way they could.